# Vector Quantization Prompting for Continual Learning

**Li Jiao[1], Qiuxia Lai[1]\*, Yu Li[2], Qiang Xu[3]**
[1] Communication University of China
[2] Harbin Institute of Technology, Shenzhen
[3] The Chinese University of Hong Kong
{jl0930,qxlai}@cuc.edu.cn;li.yu@hit.edu.cn;qxu@cse.cuhk.edu.hk

## Abstract

Continual learning requires to overcome catastrophic forgetting when training a single model on a sequence of tasks. Recent top-performing approaches are prompt-based methods that utilize a set of learnable parameters (*i.e.*, prompts) to encode task knowledge, from which appropriate ones are selected to guide the fixed pre-trained model in generating features tailored to a certain task. However, existing methods rely on predicting prompt identities for prompt selection, where the identity prediction process cannot be optimized with task loss. This limitation leads to sub-optimal prompt selection and inadequate adaptation of pre-trained features for a specific task. Previous efforts have tried to address this by directly generating prompts from input queries instead of selecting from a set of candidates. However, these prompts are continuous, which lack sufficient abstraction for task knowledge representation, making them less effective for continual learning. To address these challenges, we propose VQ-Prompt, a prompt-based continual learning method that incorporates Vector Quantization (VQ) into end-to-end training of a set of discrete prompts. In this way, VQ-Prompt can optimize the prompt selection process with task loss and meanwhile achieve effective abstraction of task knowledge for continual learning. Extensive experiments show that VQ-Prompt outperforms state-of-the-art continual learning methods across a variety of benchmarks under the challenging class-incremental setting. The code is available at `https://github.com/jiaolifengmi/VQ-Prompt`.

## 1 Introduction

Humans have the remarkable capability to continually acquire and integrate knowledge of new concepts or categories without forgetting old ones, whereas deep learning models struggle with *catastrophic forgetting* [40] when tasked with learning a sequence of classes [42, 10, 39]. Continual learning aims at addressing catastrophic forgetting in deep neural networks (DNNs) by striking a balance between plasticity for learning new incoming data effectively and stability to retain prior knowledge. Approaches in this field vary: some methods dynamically expand network architectures [64, 29, 58] or reconfigure their internal structures [48, 17, 24] for new tasks. Others penalize the update of crucial parameters from previous tasks [20, 27, 65, 1, 49] or alter parameter update rules to prevent interference across tasks [34, 7, 47, 23]. Additionally, certain methods interleave stored past data with current ones for training [18, 8, 44, 45, 4, 6, 36]. Despite recent advances, continual learning remains an open challenge for DNNs.

Recently, prompt-based continual learning has emerged as a promising solution to mitigate catastrophic forgetting in sequential task learning. This approach enhances a pre-trained Vision Trans-

---

\*Corresponding author

38th Conference on Neural Information Processing Systems (NeurIPS 2024).

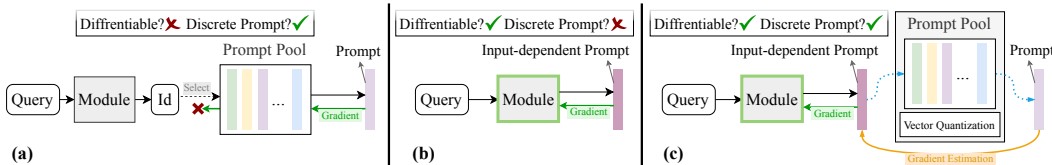

Figure 1: **Concept comparison.** (a) Prior prompt-based continual learning methods predict prompt identities for prompt selection, which cannot be optimized end-to-end with task loss. (b) Some methods enable end-to-end training by directly generating prompts from the queries using learnable parameters. However, these prompts are continuous, lacking the necessary abstraction to effectively represent the task knowledge essential for generating features tailored to a certain task. (c) Our method incorporates Vector Quantization (VQ) into the prompt generation pipeline to enable end-to-end training of discrete prompts with task loss. See §1 for details.

former (ViT) [11] with a small set of learnable parameters, known as "prompts". These prompts encapsulate task-specific knowledge, shifting the learning focus from the entire model to the prompts themselves, which guide the pre-trained model in generating task-relevant outputs. During inference, the most suitable prompt containing necessary task knowledge is selected from the prompt pool based on the input image to direct the behavior of the frozen pre-trained model.

Current prompt-based methods either involve a key-query matching mechanism to select prompts based on the similarity between the image features and the key parameters paired with prompts [62, 61], or explicitly predict the prompt indices and perform the selection accordingly [60, 59]. However, the non-differentiable nature of indexing impedes the prompt selection from being optimized end-to-end with task loss. This limitation can lead to diminished performance, as inaccurate prompt selection may fail to tailor the pre-trained features for the specific task. Efforts to address this issue include implementing a differentiable prompt selection, such as generating prompts as a weighted sum from the prompt pool [50], or deriving prompts from the intermediate features of the input image [51, 26]. However, the resulting prompts are continuous, which lack the necessary abstraction to effectively represent the task knowledge essential for guiding the pre-trained model to generate features tailored to a certain task. A concept comparison is shown in Fig. 1.

The assumption that discrete prompts better represent task knowledge than continuous prompts for continual learning can be supported by both theoretical insights from cognitive science and empirical evidence. Discrete prompts mimic the organizational structure of memory and knowledge in the human brain, which is typically understood to consist of discrete units such as concepts and facts [19]. This clear separation of information helps prevent interference among different knowledge domains, and enables models to provide distinct guidance for feature extraction specific to each task. Such knowledge abstraction aligns with categorical perception in human cognition, where sensory inputs are perceived as distinct categories (e.g., colors, phonemes) rather than continuous spectrum [41]. Furthermore, empirical comparisons in §5.2 demonstrate the effectiveness of discrete prompts when optimized end-to-end with task loss (*e.g.*, VQ-Prompt V.S. CODA-P or EvoPrompt). In summary, discrete prompts hold significant promise for improving the continual learning capabilities of models, bringing them more in line with human learning.

Optimizing prompts with task loss while preserving their discrete properties as representations of concepts poses a non-trivial challenge. In this paper, we introduce Vector Quantization Prompting (VQ-Prompt) for continual learning, which can optimize prompts using task loss while preserving their discrete characteristics as concept representations. This method involves initially generating a continuous prompt and then replacing it with its nearest match from a predefined prompt pool. To address the non-differentiability inherent in prompt quantization, we apply gradient estimation to propagate task loss to the continuous prompt, while additional vector quantization regularization terms further refine the learning of the prompt pool. To further stabilize task knowledge learning, we use representation statistics to mitigate the classification bias towards previous tasks, thereby enhancing continual learning performance.

Our contributions are three-folds: (1) We propose VQ-Prompt, an end-to-end learnable discrete prompting mechanism for continual learning, addressing a critical yet overlooked aspect in the current literature. (2) We leverage gradient estimation to pass the task loss to prompt-related parameters while regularizing the learning of the prompts with vector quantization terms, which facilitates the

end-to-end training of the discrete prompt pool. (3) We incorporate representation statistics during training to further stabilize task knowledge learning and improve the overall continual learning performance. Extensive experiments show that VQ-Prompt consistently outperforms state-of-the-art continual learning methods on a variety of benchmarks.

## 2 Related work

**Continual Learning** refers to the process where multiple tasks are learned sequentially without forgetting [42, 10, 39]. Generally, continual learning has three scenarios [54]. Task-incremental learning (TIL) learns different classes for each task and assumes having task identities available at test time. Domain-incremental learning (DIL) maintains the same set of classes for different tasks while changing the data distributions across tasks, and task identities are not provided for inference. For Class-incremental learning (CIL), each task involves new classes and all the learned classes are to be classified without task identities available during inference. In this paper, we focus on the more representative and challenging CIL scenario. Numerous efforts have been devoted to alleviating catastrophic forgetting. *Architecture-based methods* address this by either dynamically expanding network architectures [64, 29, 58] or modifying internal network structures [48, 17, 24] for new tasks. *Regularization-based methods* focus on limiting updates to vital parameters from earlier tasks [20, 27, 65, 1, 49], or modifying the rules for parameter updates to reduce task interference [34, 7, 47, 23]. *Rehearsal-based methods* incorporate previous data with current data during training to mitigate forgetting [18, 8, 44, 45, 4, 6, 36]. Despite recent advances, continual learning remains a challenging and evolving field.

**Prompt-based Continual Learning Methods.** Recently, there has been a surge in methods leveraging prompting techniques from natural language processing (NLP) [28, 30] for continual learning. These methods instruct a frozen pre-trained transformer using learnable prompts that encode task knowledge. During training, prompt selection is either through key-query similarity matching [62] or indicated by task identity [61, 12, 60, 59]. In inference, the appropriate prompt is chosen through similarity matching with key or feature centroids. However, both kinds of prompt selection are non-differentiable, making it challenging to optimize them end-to-end with the task loss, particularly when the gap between the pre-training task and unknown future tasks is large. To address this, CODA-Prompt [50] adopts a soft prompt selection, *i.e.*, generating prompts as a weighted sum from the prompt pool. APG [51] and EvoPrompt [26] learn to derive prompts from intermediate image features. Nevertheless, all three methods generate prompts that are continuous, which lack the necessary abstraction to effectively represent the task knowledge essential for instructing the pre-trained model to produce features tailored to a certain task. In this paper, we present a new prompting framework for continual learning capable of optimizing prompts with task loss while preserving their discrete properties as the representation of task knowledge.

**Vector Quantization in Representation Learning.** Vector Quantization (VQ) is a technique used in signal processing and data compression to represent a set of vectors (data points) with a smaller set of "coding vectors" (CVs). Unsupervised VQ algorithms such as Self-organizing Maps (SOMs) [22] and Neural Gas (NG) networks [38] attempt to obtain a set of CVs that optimally represent the data. Supervised VQ algorithms such as Learning Vector Quantization (LVQ) [21] focus on reducing the misclassification rates by refining decision boundaries between classes. In generative modeling, VQ has been used to learn structured discrete latent spaces in VQ-VAE [55] and VQ-GAN [13] to achieve higher fidelity images. Recently studies have explored combining VQ with continual learning to constrain the feature space, aiming to enhance class separation and retrain prior knowledge across increments [52, 53, 9, 37]. In this paper, instead of utilizing VQ to confine the feature space, we employ VQ to enable end-to-end learning a set of discrete prompts that effectively encode task knowledge in a learning system that evolves over time.

## 3 Preliminary

**Problem Formulation.** In class-incremental learning (CIL), a model is required to sequentially learn a series of tasks with disjoint class sets, and to accurately classify all seen classes during evaluation. Formally, let $\mathcal{D}_t = \{(\boldsymbol{x}_i^t, y_i^t)\}_{i=1}^{N_t}$ denote the training set of the $t$-th task, where $\boldsymbol{x}_i^t \in \mathcal{X}_t$ is an input image, $y_i^t \in \mathcal{Y}_t$ is the target label, and $N_t$ is the number of samples. The label spaces of all the tasks are mutually exclusive, *i.e.*, $\cap_{t=1}^T \mathcal{Y}_t = \emptyset$, where $T$ is the total number of tasks. Consider a deep

learning model $\mathcal{M} = \phi \circ f$ with a backbone $f(\cdot)$ and a classifier $\phi(\cdot)$. During training on task $t$, the model only has access to $\mathcal{D}_t$, which raises a risk of forgetting old tasks. After learning task $t$, the model is expected to perform well on all classes in $\mathcal{Y}_{1:t} = \cup_{k=1}^{t} \mathcal{Y}_k$, and further on $\mathcal{Y}_{1:T}$ after completing training on all $T$ tasks.

**Prompt-based Learning** is an emerging approach in NLP [32] that involves incorporating extra instructions into pre-trained models to guide their performance on specific tasks. Rather than relying on extensive retraining or task-specific fine-tuning, this technique leverages prompts to shape the behavior of pre-trained models, providing adaptable instructions that helps them handle a wide range of downstream tasks more effectively. In vision-related continual learning, prompting is typically employed with Vision Transformer (ViT) [11].

ViT consists of a sequence of multi-head self-attention (MSA) blocks [56]. For clarity, we take one MSA block as an example to illustrate the prompting. We denote the input query, key, and value of the MSA block as $\boldsymbol{h}_Q, \boldsymbol{h}_K$ and $\boldsymbol{h}_V$, respectively. Here, $\boldsymbol{h}_* \in \mathbb{R}^{L \times D}$, $L$ is the sequence length, and $D$ is the embedding dimension. The output of the MSA is computed as:

$$\text{MSA}(\boldsymbol{h}_Q, \boldsymbol{h}_K, \boldsymbol{h}_V) = \text{Concat}(\text{h}_1, \ldots, \text{h}_M)W^O,$$
$$\text{h}_m = \text{Attention}(\boldsymbol{h}_Q W_m^Q, \boldsymbol{h}_K W_m^K, \boldsymbol{h}_V W_m^V), \tag{1}$$

where $W^O, W_m^Q, W_m^K$ and $W_m^V$ are projection matrices, $m = 1, \cdots, M$ is the head index, $M$ is the number of heads, and $\boldsymbol{h}_Q = \boldsymbol{h}_K = \boldsymbol{h}_V = \boldsymbol{h}$ for SA.

Previous prompt-based continual learning methods mainly implement Prompt Tuning (Pro-T) [28] and Prefix Tuning (Pre-T) [30]. Pro-T prepends the same prompt $\boldsymbol{p} \in \mathbb{R}^{L_p \times D}$ to $\boldsymbol{h}_Q, \boldsymbol{h}_K$, and $\boldsymbol{h}_V$. The prompting function of Pro-T is defined as:

$$f_{\text{Pro-T}}(\boldsymbol{p}, \boldsymbol{h}) = \text{MSA}([\boldsymbol{p}; \boldsymbol{h}_Q], [\boldsymbol{p}; \boldsymbol{h}_K], [\boldsymbol{p}; \boldsymbol{h}_V]), \tag{2}$$

where $[\cdot; \cdot]$ means concatenating along the sequence length dimension. The output dimension is $(L_p + L) \times D$. Pre-T splits $\boldsymbol{p}$ along the sequence length dimension into $\boldsymbol{p}_K, \boldsymbol{p}_V \in \mathbb{R}^{L_p/2 \times D}$, which are prepended to $\boldsymbol{h}_K$ and $\boldsymbol{h}_V$, respectively:

$$f_{\text{Pre-T}}(\boldsymbol{p}, \boldsymbol{h}) = \text{MSA}(\boldsymbol{h}_Q, [\boldsymbol{p}_K; \boldsymbol{h}_K], [\boldsymbol{p}_V; \boldsymbol{h}_V]). \tag{3}$$

The output dimension is the same as that of $\boldsymbol{h}$. In continual learning, the pre-trained ViT backbone is kept frozen as a general feature extractor, and the prompt parameters $\boldsymbol{p}$ are trained to capture task knowledge. Proper prompts corresponding to the input samples are selected to guide the feature extraction during inference. Following [61, 50, 59], we adopt Pre-T strategy in our method.

## 4 Method

As shown in Fig. 2, our VQ-Prompt approach begins by constructing a continuous prompt through a soft selection from the prompt pool (§4.1). The continuous prompt is then quantized to an element in the prompt pool, which is inserted into an MSA block of a frozen pre-trained transformer. This process is made end-to-end trainable through gradient estimation and vector quantization regularization (§4.2), such that the prompting parameters, namely the keys and the prompt pool could all be optimized using the task loss. In this way, VQ-Prompt can yield a discrete prompt for each input while maintaining end-to-end optimization. To better stabilize task knowledge learning, representation statistics of previously learned classes are employed to mitigate the classification bias (§4.3).

### 4.1 Prompt Formation

Most previous prompting-based continual learning approaches construct their prompts by selecting from the prompt pool based on key-query similarity [62, 61] or other task identity prediction mechanisms [60, 59], making the prompt selection process non-differentiable. In our prompt formation, we first generate a continuous prompt by aggregating all the elements in the prompt pool based on the similarity scores between the query and the keys. Specifically, given a query $\boldsymbol{q}$ from the input image, the similarity score is calculated as:

$$\alpha = \text{Softmax}(\boldsymbol{K}\boldsymbol{q}), \tag{4}$$

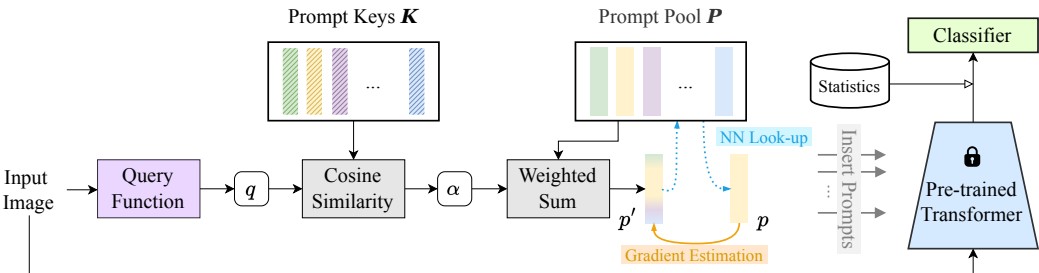

Figure 2: **VQ-Prompt framework.** An input image is passed through a query function (*e.g.*, a fixed pre-trained ViT) to generate a query $\boldsymbol{q}$, which is then used to compute similarity scores with prompt keys $\boldsymbol{K}$. These scores $\alpha$ serve as weights to aggregate elements from the prompt pool $\boldsymbol{P}$ to form a continuous prompt $\boldsymbol{p}'$. This prompt is subsequently quantized to an element within the prompt pool $\boldsymbol{p}$, and then fed into a specific MSA block of a frozen pre-trained transformer. To ensure differentiability, the prompt quantization process employs gradient estimation and prompt pool regularization. The representation statistics of features from learned classes are used to stabilize task knowledge learning. More details are shown in §4.

where $\boldsymbol{K} \in \mathbb{R}^{N \times D}$ is the prompt key matrix, $\boldsymbol{q} \in \mathbb{R}^D$ is the query, $N$ is the number of keys, and $D$ is the embedding dimension. Then, the continuous prompt is obtained by:

$$\boldsymbol{p}' = \sum_i \alpha_i \boldsymbol{P}_i, \quad i = 1, \cdots, N, \tag{5}$$

where $\boldsymbol{P}_i \in \mathbb{R}^{L_p \times D}$ is the $i$-th element in the prompt pool, and $L_p$ is the length of the prompt. Such a prompt formation process is differentiable and can be viewed as a simplified version of CODA-P [50]. Here, we do not learn an extra attention parameter for weighting the query, nor do we increase the number of elements in the prompt pool or the number of keys during sequential task learning.

### 4.2 Vector Quantization Prompting (VQ-Prompt)

**Nearest-neighbour Look-up.** The continuous prompt $\boldsymbol{p}'$ obtained in Eq. (5) is conditioned on a specific instance, *i.e.*, it varies with the input images, making it insufficiently abstract to capture task knowledge effectively. We further perform prompt quantization by performing the nearest neighbour (NN) look-up in the prompt pool $\boldsymbol{P}$ using $\boldsymbol{p}'$. The quantized prompt to be fed to the MSA block is obtained by:

$$\boldsymbol{p} = \boldsymbol{P}_k, \;\; k = \arg\min_j \|\boldsymbol{p}' - \boldsymbol{P}_j\|_2, \;\; \boldsymbol{p} \in \mathbb{R}^{L_p \times D}. \tag{6}$$

Such a prompt selection pipeline can be viewed as a specific non-linearity that maps the continuous prompt to 1-of-$N$ elements in the prompt pool.

**Gradient Estimation.** Because the $\arg\min$ operation in Eq. (6) is non-differentiable, we use the straight-through estimator [3] to approximate the gradient of $\boldsymbol{p}'$ using the gradient of $\boldsymbol{p}$. Despite its simplicity, this estimator has demonstrated its effectiveness in our experiments. Specifically, in the forward process, the quantized prompt $\boldsymbol{p}$ is passed to the MSA block in the pre-trained transformer. During the backward computation, the gradient of $\boldsymbol{p}$ is transferred unaltered to $\boldsymbol{p}'$, and the optimization of prompt pool $\boldsymbol{P}$ and keys $\boldsymbol{K}$ guided by the similarity scores (*c.f.*, Eq. (4)). This gradient estimation is justified, as $\boldsymbol{p}$ and $\boldsymbol{p}'$ share the same $L_p \times D$-dimensional space, and the gradient of $\boldsymbol{p}$ provides valuable information on how prompt parameter learning could instruct the transformer features to minimize the cross-entropy (CE) loss during task learning. In this way, each prompt and key element is adjusted according to its relevance to the current learning context, rather than undergoing wholesale changes. This allows for more updates of task-relevant elements without disrupting less relevant ones, thereby maintaining previously acquired knowledge while adapting to new tasks.

**Vector Quantization (VQ) Regularization.** Though the prompt pool $\boldsymbol{P}$ could receive gradients from the task loss through straight-through gradient estimation of mapping from $\boldsymbol{p}$ to $\boldsymbol{p}'$, to enhance the learning of the prompt embedding space, we add an extra VQ objective. This VQ objective uses

the $L_2$ error to move the selected element $p$ of prompt pool towards the continuous prompt $p'$:

$$\mathcal{L}_{\text{VQ}} = \|\text{sg}[p'] - p\|_2^2. \tag{7}$$

Here, $\text{sg}[\cdot]$ stands for the stop-gradient operation [55], which constrains its operand to be a non-updated constant during training.

To ensure that the learning processes of the prompt keys $K$ and the continuous prompt $p'$ align closely with the characteristics of the element $p$ from the prompt pool $P$, we further introduce a commitment regularization term. This term is defined mathematically as follows:

$$\mathcal{L}_{\text{Commit}} = \|p' - \text{sg}[p]\|_2^2. \tag{8}$$

The incorporation of this commitment loss ensures that the prompt formation process described in §4.1 is optimized to yield prompts to commit to the elements in the prompt pool $P$, thereby promoting consistency and stability in the prompt learning process.

### 4.3 Stabilizing Task Knowledge Learning with Representation Statistics

Though prompts effectively capture task knowledge to guide the backbone $f(\cdot)$ in producing instructed representations, the classifier $\phi(\cdot)$ may develop a bias towards new classes in continual learning scenarios [16, 2]. This bias can adversely affect the learning of the prompts for subsequent tasks. To mitigate this issue and stabilize task knowledge learning, we employ a strategy similar to [59], which leverages the representation statistics of previously learned classes to correct classifier bias and stabilize prompt learning. Specifically, after completing task $t$, we calculate the mean $\mu_c$ and variance $\sigma_c$ for each class $c \in \mathcal{Y}_{1:t}$ with the learned prompt parameters and the pre-trained backbone. By modeling each class as a Gaussian distribution, we generate pseudo features through sampling from these distributions. These pseudo features are then used to fine-tune the classifier, thereby mitigating its bias towards recent classes. The balanced classifier could help stabilize task knowledge learning in the prompts, alleviate catastrophic forgetting, and enhance overall continual learning performance.

### 4.4 Overall Optimization Objective

The overall loss function is defined in Eq. (9), which extends the task loss $\mathcal{L}_{\text{CE}}$ with two terms, namely a quantization objective $\mathcal{L}_{\text{VQ}}$ weighted by $\lambda_q$, and a commitment term $\mathcal{L}_{\text{Commit}}$ weighted by $\lambda_c$.

$$\mathcal{L} = \mathcal{L}_{\text{CE}} + \lambda_q \mathcal{L}_{\text{VQ}} + \lambda_c \mathcal{L}_{\text{Commit}}. \tag{9}$$

Here, $\mathcal{L}_{\text{CE}}$ is the cross-entropy loss that supervises the learning of the image classification task, $\mathcal{L}_{\text{VQ}}$ is the VQ regularization defined in Eq. (7) that updates the prompt pool elements to move towards the continuous prompt $p'$, and $\mathcal{L}_{\text{Commit}}$ is the commitment term defined in Eq. (8) that forces the prompt formation process to commit to the prompt pool elements. During training, the pre-trained backbone $f(\cdot)$ is frozen, while the classifier $\phi(\cdot)$ and prompting parameters $K$ and $P$ are optimized across all the tasks. During task knowledge stabilization, only the classifier $\phi(\cdot)$ is actively trained.

## 5 Experiment

### 5.1 Experimental Setups

**Datasets.** We consider three representative benchmarks for evaluating CIL. *ImageNet-R* [14] includes 200-class images that are either hard samples for ImageNet or newly collected data with different styles, thus can serve as a challenging benchmark for continual learning with pre-trained models. In the experiments, we divide it into 5, 10, and 20 disjoint tasks and report the corresponding performance. *Split CIFAR-100* randomly splits the original CIFAR-100 [25] into 10 disjoint tasks, each containing 10 classes. *Split CUB-200* is built on CUB-200-2011 [57], a fine-grained classification dataset, by randomly splitting the 200 classes into 10 tasks, where each task contains 20 classes.

**Baselines.** We evaluate our approach against a comprehensive set of baselines to contextualize its performance. Following [50], we include "Joint Training" as the upper-bound performance, setting a benchmark for optimal results. To establish lower bounds, we employ two sequential learning baselines, denoted as "FT" and "FT++", with the latter refraining from updating the logits of previously learned classes during the training of new tasks.

We also consider 5 prompt-based approaches: L2P [62], DualPrompt [61], HiDe-Prompt [59], CODA-Prompt [50] and EvoPrompt [26], where the last two yield continuous prompts. For L2P, we include

Table 1: **Comparison on ImageNet-R.** Results on "5-task", "10-task", and "20-task" settings are included. Backbones are pre-trained on ImageNet-1K. ↑ denotes larger values are better. See §5.2.

| Method | Pub. | 5-task | | 10-task | | 20-task | |
|---|---|---|---|---|---|---|---|
| | | FAA (↑) | CAA (↑) | FAA (↑) | CAA (↑) | FAA (↑) | CAA (↑) |
| Joint-Train. | | 82.06 | | 82.06 | | 82.06 | |
| FT | | $18.74 \pm 0.44$ | $48.39 \pm 0.58$ | $10.12 \pm 0.51$ | $35.23 \pm 0.92$ | $4.75 \pm 0.40$ | $22.8 \pm 0.37$ |
| FT++ | | $60.42 \pm 0.87$ | $71.59 \pm 0.50$ | $48.93 \pm 1.15$ | $66.79 \pm 0.92$ | $35.98 \pm 1.38$ | $59.68 \pm 0.95$ |
| L2P++ [62] | CVPR22 | $70.83 \pm 0.58$ | $78.34 \pm 0.47$ | $69.29 \pm 0.73$ | $78.30 \pm 0.69$ | $65.89 \pm 1.30$ | $77.15 \pm 0.65$ |
| Deep L2P++ [62] | CVPR22 | $73.93 \pm 0.37$ | $80.14 \pm 0.54$ | $71.66 \pm 0.64$ | $79.63 \pm 0.90$ | $68.42 \pm 1.20$ | $78.68 \pm 1.03$ |
| DualPrompt [61] | ECCV22 | $73.05 \pm 0.50$ | $79.47 \pm 0.40$ | $71.32 \pm 0.62$ | $78.94 \pm 0.72$ | $67.87 \pm 1.39$ | $77.42 \pm 0.80$ |
| CODA-P [50] | CVPR23 | $76.51 \pm 0.38$ | $82.04 \pm 0.54$ | $75.45 \pm 0.56$ | $81.59 \pm 0.82$ | $72.37 \pm 1.19$ | $79.88 \pm 1.06$ |
| HiDe-Prompt* [59] | NeurIPS23 | $76.29 \pm 0.10$ | $78.77 \pm 0.11$ | $76.74 \pm 0.18$ | $78.76 \pm 0.11$ | $76.46 \pm 0.06$ | $78.76 \pm 0.11$ |
| EvoPrompt [26] | AAAI24 | $77.16 \pm 0.18$ | $82.22 \pm 0.54$ | $76.83 \pm 0.08$ | $82.09 \pm 0.68$ | $74.41 \pm 0.23$ | $80.96 \pm 1.42$ |
| **VQ-Prompt** | **—** | $\mathbf{79.23 \pm 0.29}$ | $\mathbf{82.96 \pm 0.50}$ | $\mathbf{78.71 \pm 0.22}$ | $\mathbf{83.24 \pm 0.68}$ | $\mathbf{78.10 \pm 0.22}$ | $\mathbf{82.70 \pm 1.16}$ |

* denotes results obtained by running the official code with ImageNet-1K pre-trained weights.

Table 2: **Comparison on Split CIFAR-100.** Backbones are pre-trained on ImageNet-1K. See §5.2 for details.

| Method | Pub. | 10-task | |
|---|---|---|---|
| | | FAA (↑) | CAA (↑) |
| Joint-Train. | | 91.38 | |
| FT | | $29.21 \pm 0.18$ | $37.37 \pm 0.89$ |
| FT++ | | $49.91 \pm 0.42$ | $74.76 \pm 0.93$ |
| LwF [31] | TPAMI17 | $64.83 \pm 1.03$ | - |
| L2P++ [62] | CVPR22 | $82.50 \pm 1.10$ | $88.96 \pm 0.82$ |
| Deep L2P++ [62] | CVPR22 | $84.30 \pm 1.03$ | $90.50 \pm 0.69$ |
| DualPrompt [61] | ECCV22 | $66.00 \pm 0.57$ | $77.92 \pm 0.50$ |
| CODA-P [50] | CVPR23 | $70.03 \pm 0.47$ | $74.26 \pm 0.24$ |
| EvoPrompt [26] | AAAI24 | $87.97 \pm 0.30$ | $92.26 \pm 0.86$ |
| **VQ-Prompt** | **—** | $\mathbf{88.73 \pm 0.27}$ | $\mathbf{92.84 \pm 0.73}$ |

Table 3: **Comparison on Split CUB-200.** Backbones are pre-trained on ImageNet-21K. * denotes backbone is not frozen. See §5.2.

| Method | Pub. | 10-task | |
|---|---|---|---|
| | | FAA (↑) | CAA (↑) |
| Joint-Train. | | 88.00 | |
| FT | | $11.04 \pm 0.78$ | $31.96 \pm 0.74$ |
| FT++ | | $37.81 \pm 2.86$ | $63.55 \pm 1.62$ |
| LwF [31] | TPAMI17 | $69.75 \pm 1.37$ | $80.45 \pm 2.08$ |
| BiC [63] | CVPR19 | $81.91 \pm 2.59$ | $89.92 \pm 1.57$ |
| DualPrompt [61] | ECCV22 | $66.00 \pm 0.57$ | $77.92 \pm 0.50$ |
| CODA-P [50] | CVPR23 | $70.03 \pm 0.47$ | $74.26 \pm 0.24$ |
| *SLCA [67] | ICCV23 | $84.71 \pm 0.40$ | $\mathbf{90.94 \pm 0.68}$ |
| HiDe-Prompt [59] | NeurIPS23 | $86.61 \pm 0.18$ | $87.01 \pm 0.03$ |
| **VQ-Prompt** | **—** | $\mathbf{86.72 \pm 0.94}$ | $90.33 \pm 1.03$ |

its two variations from [50]: "L2P++" and "Deep L2P++". L2P++ uses Pre-T instead of Pro-T and inserts the prompts to the first MSA block, which achieves better performance than the original L2P [33]. Deep L2P++ extends L2P++ by incorporating prompts into the first five MSA blocks.

In addition to prompt-based methods, we include a classical regularization-based method LwF [31], and a rehearsal-based method BiC [63], providing a more comprehensive overview for evaluation.

**Evaluation Metrics.** We present *Final Average Accuracy (FAA)* and *Cumulative Average Accuracy (CAA)* for comparison. FAA refers to the last average accuracy after learning all the tasks, which is equivalent to "Last-Acc" in [67]. CAA is the average of historical FAA values after learning each task, which is equivalent to "Inc-Acc" in [67]. The formal definitions of the metrics are in §A.1.

**Implementation Details.** We follow prior works [62, 61, 50, 59, 67, 26] and use ViT-Base [11] pre-trained with supervised learning on ImageNet-1K [46] or ImageNet-21K [43] as the backbone. The number of keys and prompt elements $N$ is 10. The prompt length $L_p$ is 8. The embedding dimension $D = 768$ which is the same as the feature dimension of ViT-Base.

Our method is trained using an AdamW optimizer [35] with an initial learning rate of 0.0025 and a cosine decay schedule. The batch size is 128 for Split CIFAR-100 and Split CUB-200, and 64 for ImageNet-R. The number of epochs is set to be 20 for training on all three datasets. The classifier bias mitigation process described in §4.3 requires ten epochs of training. Each experiment is run on a single NVIDIA GeForce RTX 4090 GPU. More details are presented in §A.2.

## 5.2 Comparison Results

In this section, we present a comprehensive comparison with established baselines across various datasets and pre-training regimes. The performances of different methods are reported in separate

Table 4: Results on 10-task ImageNet-R with different self-supervised pre-training paradigms.

| Method | Pub. | iBOT-1K [68] | | DINO-1K [5] | |
|---|---|---|---|---|---|
| | | FAA (↑) | CAA (↑) | FAA (↑) | CAA (↑) |
| DualPrompt [61] | ECCV22 | $61.51 \pm 1.05$ | $67.11 \pm 0.08$ | $58.57 \pm 0.45$ | $64.89 \pm 0.15$ |
| CODA-Prompt [50] | CVPR23 | $66.56 \pm 0.68$ | $73.14 \pm 0.57$ | $63.15 \pm 0.39$ | $69.73 \pm 0.25$ |
| HiDe-Prompt [59] | NeurIPS23 | $71.33 \pm 0.21$ | $73.62 \pm 0.13$ | $68.11 \pm 0.18$ | $71.70 \pm 0.01$ |
| **VQ-Prompt** | **—** | $\mathbf{71.68 \pm 0.72}$ | $\mathbf{76.66 \pm 0.40}$ | $\mathbf{68.42 \pm 0.28}$ | $\mathbf{74.43 \pm 0.58}$ |

tables due to the varying experimental settings such as the number of tasks and the pre-training dataset, to ensure a fair and accurate comparison.

**Results on ImageNet-R.** Table 1 shows the results across five runs on ImageNet-R with 5-task, 10-task, and 20-task splits using the ViT-Base backbone pre-trained with supervised learning on ImageNet-1K. Our VQ-Prompt consistently outperforms other methods across key metrics such as FAA and CAA for all task splits, including the latest prompt-based method EvoPrompt.

**Results on Split CIFAR-100.** Table 2 presents the results across five runs on CIFAR-100 split into 10 tasks, with the ViT-Base backbone also pre-trained on ImageNet-1K with supervised learning. Our VQ-Prompt achieves superior results compared to other methods using the same pre-training weights.

**Results on Split CUB-200.** Table 3 displays the results on Split CUB-200. Following [67], we use the ViT-Base backbone pre-trained on ImageNet-21K for this dataset. VQ-Prompt achieves superior or comparable performance compared with all other methods, including SLCA [67], which trains the entire network without freezing the pre-trained feature extraction backbone. This highlights its potential and efficacy in continual learning for fine-grained classification tasks.

**Other Pre-training Regimes.** Table 4 summarizes the experimental results across three runs on the 10-task ImageNet-R dataset, utilizing different self-supervised pre-training paradigms, namely iBOT-1K [68] and DINO-1K [5]. These results demonstrate that our method consistently outperforms state-of-the-art prompt-based continual learning methods, underscoring its robustness and efficiency in leveraging self-supervised pre-training for continual learning tasks. Specifically, VQ-Prompt shows a greater advantage in CAA than FAA, indicating its superior ability to leverage past knowledge for aiding current tasks, despite a slightly higher degree of forgetting relative to some baselines. This trade-off between adaptation to new tasks and forgetting of previous ones is a common challenge in continual learning. With self-supervised pre-training, our method tends to prioritize adaptability to new tasks to ensure that the model remains relevant and effective in dynamic environments.

## 5.3 Ablation Study and Additional Analysis

In this section, we assess the effectiveness of different components illustrated in §4. The experiments are performed on 10-task ImageNet-R with the ViT-Base backbone pre-trained on ImageNet-1K.

**Effectiveness of VQ Design.** Our VQ design (*c.f.*, §4.2) enables end-to-end training of the discrete prompt selection in continual learning. Here, we compare it with an alternative intuition design choice, *i.e.*, rewriting Eq. (4) as $\alpha = \text{Softmax}(\boldsymbol{K}\boldsymbol{q}/\tau)$, and reducing the temperature $\tau$ of the softmax operation. A lower temperature leads to a "sharper" distribution of $\alpha$, allowing the prompt formation in Eq. (5) to more closely approximate discrete prompt selection during end-to-end training with task loss. This baseline is denoted as "Soft-Prompt". Fig. 3 (a) presents the FAA values of Soft-Prompt with different $\tau$ values. Surprisingly, Soft-Prompt achieves its best performance of 77.15 at $\tau = 1.0$ instead of at lower values. While its performance is comparable to other prompt-based methods, Soft-Prompt falls short of "VQ-Prompt-S", which achieves an FAA value of 78.05 on 10-task ImageNet-R. Here, VQ-Prompt-S is a simplified version of VQ-Prompt that does not use representation statistics. Our standard version VQ-Prompt further achieves an FAA value of 78.83. This observation underscores the effectiveness of our VQ design compared to the intuitive low-temperature soft prompt selection.

The rationale behind this is that reducing the temperature makes the softmax operation more sensitive to differences in logits. While this heightened sensitivity is acceptable when the model is confident in its predictions, it can lead to more aggressive prompt choices at the beginning of the training when the model is less fully trained. In contrast, our VQ-Prompt utilizes a standard softmax for prompt formation, substitutes the resulting prompt with the nearest one in the prompt pool, and enables

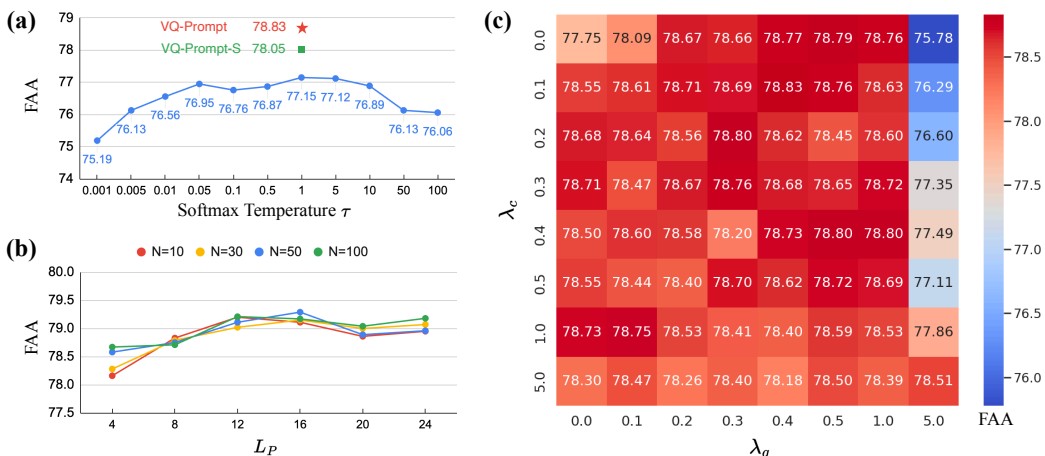

Figure 3: **Ablation study. (a) VQ Design.** We show the performance of an alternative of VQ Design, "Soft-Prompt", that generates the continuous prompt with low-temperature softmax operation only without using VQ. Here, "VQ-Prompt-S" is a simplified version of VQ-Prompt without using representation statistics. **(b) Prompt Hyperparameters.** The results of varying the size of the prompt pool $N$ and the length of a single prompt $L_p$ are displayed. **(c) Loss Weights.** The results of different combinations of $\lambda_q$ and $\lambda_c$ values are presented. See §5.3 for details.

Table 5: **Effectiveness of classifier bias mitigation.** Results for "5-task", "10-task", and "20-task" settings on ImageNet-R are included. "C.B.M." denotes "Classifier Bias Mitigation". Backbones are pre-trained on ImageNet-1K. ↑ denotes larger values are better. See §5.3 for details.

| Method | C.B.M. | 5-task | | 10-task | | 20-task | |
|---|---|---|---|---|---|---|---|
| | | FAA (↑) | CAA (↑) | FAA (↑) | CAA (↑) | FAA (↑) | CAA (↑) |
| L2P++ [62] | No | $70.83 \pm 0.58$ | $78.34 \pm 0.47$ | $69.29 \pm 0.73$ | $78.30 \pm 0.69$ | $65.89 \pm 1.30$ | $77.15 \pm 0.65$ |
| L2P++ V2 [62] | Yes | $74.11 \pm 0.08$ | $78.44 \pm 0.63$ | $72.93 \pm 0.27$ | $78.63 \pm 0.80$ | $70.99 \pm 0.26$ | $77.65 \pm 0.79$ |
| EvoPrompt [26] | No | $77.16 \pm 0.18$ | $82.22 \pm 0.54$ | $76.83 \pm 0.08$ | $82.09 \pm 0.68$ | $74.41 \pm 0.23$ | $80.96 \pm 1.42$ |
| **VQ-Prompt-S** | **No** | $\mathbf{78.52 \pm 0.34}$ | $\mathbf{82.64 \pm 0.68}$ | $\mathbf{78.00 \pm 0.39}$ | $\mathbf{82.83 \pm 0.69}$ | $\mathbf{76.19 \pm 0.26}$ | $\mathbf{81.68 \pm 1.02}$ |
| **VQ-Prompt** | **Yes** | $\mathbf{79.23 \pm 0.29}$ | $\mathbf{82.96 \pm 0.50}$ | $\mathbf{78.71 \pm 0.22}$ | $\mathbf{83.24 \pm 0.68}$ | $\mathbf{78.10 \pm 0.22}$ | $\mathbf{82.70 \pm 1.16}$ |

end-to-end learning through gradient estimation and VQ regularization, which proves to be more robust for task knowledge learning compared with Soft-Prompt.

**Hyperparameters for Prompting.** There are two key hyperparameters: i) the size of the prompt pool $N$ that represents the total capacity of the learnable prompts, and ii) the length of a single prompt $L_p$ which determines the capacity of a single prompt to encode certain aspects of task knowledge. The total size of the prompts to be prepended to the input of one MSA block is given by $N \times L_p$.

Fig. 3 (b) illustrates the impact of varying $L_p$ and $N$ on FAA performance. Across different parameter configurations, our method consistently outperforms existing approaches, demonstrating its robustness. Specifically, an excessively small $L_p$ value consistently yields sub-optimal results, as indicated by the lower FAA scores across different $N$ values. Increasing $N$ can partially compensate for small $L_p$ values, leading to improved performance. In contrast, increasing $L_p$ generally enhances performance up to a certain threshold, beyond which an overly large $L_p$ may cause knowledge overfitting, as reflected by the stable or slightly declining FAA scores for larger $L_p$ values. We selected $L_p = 8$ and $N = 10$ as our default configuration. This configuration achieves superior results with fewer parameters compared to other prompt-based methods (*c.f.*, § A.3).

Our competitive performance is primarily attributed to the use of VQ, which offers several key benefits for prompt-based continual learning. First, VQ enables the encoding of task knowledge into discrete prompts, which provide a more compact representation than continuous prompts. This discrete nature helps in capturing essential task-specific features with the necessary level of abstraction. Second, integrating VQ within the prompt-based framework facilitates end-to-end optimization with task loss, ensuring that the selected prompts are highly relevant to the task at hand, thereby enhancing the task

knowledge learning of the prompts. This enables the use of shorter prompts while maintaining strong performance, making our approach more parameter-efficient and effective.

**Impact of $\lambda_q$ and $\lambda_c$.** To further enhance the learning of prompt-related parameters, we introduce two regularization terms $\mathcal{L}_{\text{VQ}}$ and $\mathcal{L}_{\text{Commit}}$ to guide the learning (*c.f.*, §4.2). We investigate the impact of various loss weights, as shown in Eq. (9). The outcomes are detailed in Fig. 3 (c). As can be observed, these two terms can contribute to good performance when assigned with a relatively broad range of values. According to Fig. 3 (c), we set $\lambda_q = 0.4$ and $\lambda_c = 0.1$ in all of our experiments.

**Effectiveness of Classifier Bias Mitigation.** The classifier bias mitigation utilizes representation statistics to stabilize task knowledge learning (*c.f.*, §4.3), which can also be applied to other methods. To evaluate its potential advantage, we integrated this component into L2P++. As shown in Table 5, L2P++ with representation statistics ("L2P++ V2") achieves improved performance over the original L2P++ across all three ImageNet-R settings, but remains inferior to our method. Additionally, we include the results of "VQ-Prompt-S", a simplified version of VQ-Prompt that omits classifier bias mitigation. Notably, VQ-Prompt-S still outperforms other methods such as EvoPrompt, demonstrating the effectiveness of our approach. This indicates that the classifier bias mitigation process can contribute to performance improvements, but is not the sole determinant of the final performance.

## 6 Discussion and Conclusion

This study focuses on one critical deficiency inherent in current prompt-based continual learning methodologies, specifically the end-to-end optimization of the prompt selection process with task loss while keeping its discrete nature as the representation of task knowledge. Our proposed Vector Quantization Prompting (VQ-Prompt) framework mitigates the challenge by substituting continuous prompts with their nearest counterparts from the prompt pool, thereby enhancing task accuracy through a more aligned and abstract representation of conceptual task knowledge. To overcome the non-differentiability inherent in this process, we employed gradient estimation along with vector quantization regularization terms, which allows for optimizing prompt retrieval with task loss. Representation statistics are utilized to further stabilize task knowledge learning. Extensive experiments in class-incremental scenarios consistently demonstrate VQ-Prompt's superiority over SOTA methods.

**Limitations and Future Work.** One limitation of VQ-Prompt is its dependence on pre-trained models. While these models offer rich initial knowledge, enabling a more mature learning process akin to that of an adult, they also inherit the limitations of the pre-trained data distribution and the high computational costs associated with their use. This challenge is not unique to VQ-Prompt but applies broadly to other continual learning methods that rely on pre-trained models. Another limitation is the absence of constraints in calculating similarity scores and prompt keys, which can result in suboptimal prompt utilization, *i.e.*, some prompts are more frequently selected for samples from different tasks while others are less frequently used. To alleviate this, one possible strategy is to introduce constraints on prompt selection, such as limiting the reuse of prompts that have already been heavily utilized by previous tasks to enhance the diversity and utility of the prompts. We leave a more thorough exploration of such prompt selection constraints for future research.

## Broader Impacts

This paper presents a prompt-based continual learning method to tackle the challenges of knowledge preservation in sequential task learning. Our approach facilitates continual adaptation and learning, thereby contributing to the advancement of intelligent, adaptive, and efficient technologies applicable to various domains, including autonomous vehicles and personalized AI agents. While VQ-Prompt advances class-incremental continual learning, its robustness could be compromised if poisonous samples are introduced after a concept has been learned. One possible mitigation strategy is to implement robust anomaly detection methods to identify and filter out suspicious input samples before they are introduced into the training process.

## Acknowledgments and Disclosure of Funding

This work is supported by the National Natural Science Foundation of China (No. 62176241) and the Fundamental Research Funds for the Central Universities (No. CUC24QT06).

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

# A Appendix / supplemental material

In this section, we provide detailed supplementary information. §A.1 outlines the evaluation metrics used to assess performance, providing detailed descriptions and formulas for clarity. §A.3 elaborates on the configurations of prompt-based methods compared in our experiments. §A.2 offers additional implementation details, including model training procedures, detailed parameter settings, and methodology for obtaining results of other methods compared in the experiments. Finally, §A.4 provides more results on two challenging datasets.

## A.1 Evaluation Metrics

To assess the performance of continual learning, we record the average classification accuracy of all seen classes at the end of each task training, and denote the average accuracy on the $i$-th task after learning the $j$-th task as $\boldsymbol{A}_{ij}$. The formal definitions of FAA and CAA are introduced as follows.

i) *Final Average Accuracy (FAA)* refers to the last average accuracy after learning all the tasks:

$$\text{FAA} = \frac{1}{T}\sum_{i=1}^{T}\boldsymbol{A}_{iT}, \tag{10}$$

where $\boldsymbol{A}_{iT}$ is the average accuracy of task $i$ after learning task $T$, and $T$ is the number of tasks. Larger FAA indicates greater learning capacity and less forgetting. FAA is also denoted as "Last-Acc".

ii) *Cumulative Average Accuracy (CAA)* is the average of historical FAA values after learning each task, which is calculated as:

$$\text{CAA} = \frac{1}{T}\sum_{j=1}^{T}\frac{1}{j}\sum_{i=1}^{j}\boldsymbol{A}_{ij}. \tag{11}$$

CAA reflects the overall performance after learning each incremental task, which can also be denoted as "Inc-Acc".

## A.2 More Implementation Details

We use AdamW [35] with $\beta_1 = 0.9$ and $\beta_2 = 0.999$. Our batch size is 64 for ImageNet-R, and 128 for Split CIFAR-100 and Split CUB-200. We resize the input images to $224 \times 224$ and perform data transform following [50], including random horizontal flip and normalization.

Following DualPrompt [61] and CODA-Prompt [50], we use 20% of the training data as validation data, and perform hyperparameters tuning on it. After hyperparameter searching, we use a learning rate of 0.0025 for our method. For all other prompt-based methods, we use the hyperparameters following [50]

We search the values of prompt length $L_P$ from 4 to 24 with a step of 4. We search the number of prompt elements $N$ in $\{10, 30, 50, 100\}$. We found that a prompt length of 8 and 10 prompt elements already work fine. We insert prompts at the same locations as all other implemented prompt-based methods in this paper, namely, the first 5 MSA blocks. A detailed comparison of the prompt configurations can be found in §A.3.

Finally, we run FT, FT++, L2P++, Deep L2P++, DualPrompt, and CODA-Prompt by using the official implementation provided by CODA-Prompt [50]. We set the predicted logits for past task classes to be 0 to prevent gradients from flowing to the linear heads of these classes. This is recommended by CODA-Prompt to improve the performance of these methods during code reproduction, as it could alleviate the bias towards new classes in CIL for rehearsal-free methods. For HiDe-Prompt [59] and EvoPrompt [26], we reproduce the results using their respective official implementations.

## A.3 Configurations of Prompt-based Methods

Table 6 presents the configurations of all the prompt-based continual learning methods compared in our experiment. Here, "Pro-T" denotes Prompt Tuning [28], and "Pre-T" denotes Prefix Tuning strategy [30], "Locations" indicates the MSA blocks to insert the prompts, $N$ is the number of prompts/components in the prompt pool, and $L_p$ is the length of a single prompt/component. L2P++

Table 6: **Prompt configurations** for prompt-based approaches in our experiments. See §A.3.

| Approaches | Strategy | Locations | Datasets | Hyperparameters |
|---|---|---|---|---|
| L2P++ [62] | Pre-T | [0] | All | $N=30$, $L_p=20$ |
| Deep L2P++ [62] | Pre-T | [0 1 2 3 4] | All | $N=30$, $L_p=20$ |
| Dual-Prompt [61] | Pre-T | [0 1] | All | G: $N=1$, $L_p=6$ |
|  | Pre-T | [2 3 4] | All | E: $N=10$, $L_p=20$ |
| CODA-P [50] | Pre-T | [0 1 2 3 4] | All | $N=100$, $L_p=8$ |
| HiDe-Prompt [59] | Pre-T | [0 1 2 3 4] | ImageNet-R | $N=10$, $L_p=40$ |
|  |  |  | Split CIFAR-100 | $N=10$, $L_p=10$ |
|  |  |  | Split CUB-200 | $N=10$, $L_p=40$ |
| EvoPrompt [26] | Pro-T | [0 1 2 3 4 5 6 7 8 9 10 11] | All | Input-cond., $L_p=5$ |
| VQ-Prompt (Ours) | Pre-T | [0 1 2 3 4] | All | $N=10$, $L_p=8$ |

and Deep L2P++ are two variants of L2P for fair comparison. Specifically, L2P++ uses Pre-T instead of Pro-T prompting, and inserts the prompts to the first MSA block. Deep L2P++ is an extension of L2P++ with prompts incorporated into the same 5 MSA blocks as DualPrompt. For DualPrompt, "G" denotes the general prompt shared by all the tasks, and "E" denotes the expert prompt pool where only one of the elements is selected for a certain query. As can be observed, our method requires *fewer* prompting parameters while consistently achieving superior or comparable performance across the benchmarks in continual learning.

## A.4  More Experiment Results

This section presents results on two challenging datasets, namely ImageNet-A [15] and VTAB [66], for evaluating continual learning methods based on pre-trained models. ImageNet-A contains adversarial images that fool current ImageNet pre-trained classifiers, while VTAB includes 19 datasets with diverse classes that do not overlap with ImageNet-1K. For ImageNet-A, we split the 200 classes into 20 tasks. For VTAB, we sample five 10-class datasets from it to construct the cross-domain CIL setting. We used a batch size of 64 for ImageNet-A and 8 for VTAB., with other training hyperparameters consistent with those used on other datasets. As shown in Table 7, our VQ-Prompt outperforms other SOTA methods such as HiDe-Prompt when evaluated using FAA.

Table 7: Results evaluated using the FAA metric on the ImageNet-A and VTAB datasets. Backbones are pre-trained on ImageNet-1K. Larger values are better.

| Method | ImageNet-A [15] | VTAB [66] |
|---|---|---|
| HiDe-Prompt [59] | 51.67 | 86.38 |
| **VQ-Prompt** | **52.96** | **90.46** |

