# OpenReview forum: "Vector Quantization Prompting for Continual Learning"
_NeurIPS.cc/2024/Conference — NeurIPS 2024 poster_

### Official Review · Reviewer_EHaW · 2024-06-30

**Soundness:** 3
**Presentation:** 3
**Contribution:** 3
**Rating:** 5
**Confidence:** 5

**Summary:**

This paper presents VQ-Prompt, a prompt-based continual learning method using Vector Quantization (VQ) to enhance task knowledge representation and overcome catastrophic forgetting. VQ-Prompt incorporates VQ into the end-to-end training of discrete prompts, optimizing the prompt selection process with task loss and achieving effective task knowledge abstraction. Extensive experiments demonstrate that VQ-Prompt outperforms state-of-the-art methods in various benchmarks, particularly in class-incremental settings.

**Strengths:**

1. This paper is well-written and easy to understand.

2. Introducing VQ to the prompt CL is intuitive and can end-to-end train these prompts.

3. Experiments have demonstrated the effectiveness of the proposed method.

**Weaknesses:**

1. There is no constraint when calculating the similarity score \alpha and prompt key K, which may lead to corrupt prompt learning, i.e., for most test samples from different tasks, the selected prompts are similar, and large parts of prompts are useless.

2. Lack of ablation study on the fine-tuned classifier. Fine-tune classifier is a trick to improve the classifier performance, which can also be applied to other methods. It is unfair to directly compare with other methods.

**Questions:**

1. In Fig. 3(a), temperature $\tau=1$ achieves the best performance, then what about the temperature larger than 1?

---

> ### Author Rebuttal · Authors · 2024-08-06
>
> We sincerely thank the reviewer for the valuable and insightful comments.
>
> **W1. There is no constraint when calculating the similarity score \alpha and prompt key K, which may lead to corrupt prompt learning, i.e., for most test samples from different tasks, the selected prompts are similar, and large parts of prompts are useless.**
>
> Thanks for the insightful comment. We have observed that at certain layers, several prompts are more frequently selected for test samples from different tasks, while others are less frequently used. To alleviate this, one possible strategy is to introduce constraints for prompt selection, such as excluding prompts that have already been frequently used by previously learned tasks to enhance the diversity and utility of the prompts. We will explore it as future work.
>
> **W2. Lack of ablation study on the fine-tuned classifier. Fine-tune classifier is a trick to improve the classifier performance, which can also be applied to other methods. It is unfair to directly compare with other methods.**
>
> In Section 5.3, we show the performance of “VQ-Prompt-S”, a simplified version of VQ-Prompt that does not use representation statistics to mitigate the negative effect of classifier bias developed during continual learning. VQ-Prompt-S achieves an FAA value of 78.05 (single run) on 10-task ImageNet-R, which is superior than the second best method EvoPrompt with an FAA of 77.16.
>
> To address your concerns, we trained another method L2P++ with this component. The results are shown in the following table. As can be observed, L2P++ with representation statistics (“L2P++ V2”) performs better than the original L2P ++ in the “5-task”, “10-task”, and “20-task” settings on ImageNet-R, but remains inferior to our method. Further, our VQ-Prompt-S outperforms other methods such as EvoPrompt (AAAI24), demonstrating the effectiveness of our approach. The results show that while representation statistics can contribute to performance improvements,  they are not the sole determinant of the final performance.
>
> ||5-task FAA|5-task CAA|10-task FAA|10-task CAA|20-task FAA|20-task CAA|
> |-|-|-|-|-|-|-|
> | L2P++       | 70.83 $\pm$ 0.58 | 78.34 $\pm$ 0.47 | 69.29 $\pm$ 0.73 | 78.30 $\pm$ 0.69 | 65.89 $\pm$ 1.30 | 77.15 $\pm$ 0.65 |
> | L2P++ V2    | 74.11 $\pm$ 0.08 | 78.44 $\pm$ 0.63 | 72.93 $\pm$ 0.27 | 78.63 $\pm$ 0.80 | 70.99 $\pm$ 0.26 | 77.65 $\pm$ 0.79 |
> | EvoPrompt   | 77.16 $\pm$ 0.18 | 82.22 $\pm$ 0.54 | 76.83 $\pm$ 0.08 | 82.09 $\pm$ 0.68 | 74.41 $\pm$ 0.23 | 80.96 $\pm$ 1.42 |
> | VQ-Prompt-S | 78.52 $\pm$ 0.34 | 82.64 $\pm$ 0.68 | 78.00 $\pm$ 0.39 | 82.83 $\pm$ 0.69 | 76.19 $\pm$ 0.26 | 81.68 $\pm$ 1.02 |
> | VQ-Prompt   | 79.32 $\pm$ 0.29 | 82.96 $\pm$ 0.50 | 78.71 $\pm$ 0.22 | 83.24 $\pm$ 0.68 | 78.10 $\pm$ 0.22 | 82.70 $\pm$ 1.16 |
>
> **Q1.  In Fig. 3(a), temperature $\tau=1$ achieves the best performance, then what about the temperature larger than 1?**
>
> Following the suggestion, we conducted additional experiments regarding temperatures larger than 1. The results are shown in the following table. As can be observed, $\tau=5$ achieves a result comparable to $\tau=1$. However, as $\tau$ increases, the performance decreases. These findings indicate that a temperature of 1 yields the best performance for our method.
>
> |$\tau=1$|$\tau=5$|$\tau=10$|$\tau=50$|$\tau=100$|
> |-|-|-|-|-|
> |77.15|77.12|76.89|76.13|76.06|

---

> > ### Comment · Reviewer_EHaW · 2024-08-13
> > **Thanks for the rebuttal**
> >
> > I thank the authors' answers to my question. After going through all the other reviews and the given rebuttal, I will keep my score to 5 because I'm still concerned the unconstrained prompt selection may be similar, leading to wasted prompts.

---

### Official Review · Reviewer_GsNc · 2024-07-09

**Soundness:** 4
**Presentation:** 4
**Contribution:** 4
**Rating:** 7
**Confidence:** 5

**Summary:**

The representations learned by large pre-trained models have led to many Continual Learning approaches based on these models. Specifically, prompt-based approaches train a small set of learnable parameters (prompts) to guide a fixed pre-trained model for a particular task. One key component in these approaches is the selection of relevant prompts, which helps to specialise the guidance depending on the need for the corresponding input. Most previous works select the relevant prompt based on the similarity of the input and a key in the representation space of the same visual pre-trained model. The key is commonly connected to the prompt via a key-value data structure, making it unfeasible to traceable optimisation sequence and infeasible to optimise with task loss. In this paper, the authors propose Vector Quantization Prompting, which incorporates vector quantisation into an end-to-end training process, including the keys and prompts. By using a look-up NN over a weighted sum of all the prompts and a straight-through estimator to approximate the gradient for the prompts and keys, the authors were able to optimise not only the prompts but also the key when selecting relevant prompts, which, as shown in the results, can help achieve better performance in diverse benchmarks.

**Strengths:**

- The authors motivate the proposal by pointing out a clear disadvantage of current prompt-based methods: the unoptimised selection of prompts. This paper presents a creative and simple approach to tackling this issue, which also performs better, as shown in the results.
- The paper is well-written and structured to help readers understand the problem and the authors' proposed solution. Although the solutions lack some intuitions and explanations concerning the reasons for the comparisons and ablations, this could be because of the lack of space.

**Weaknesses:**

- One of the authors' motivations is learning discrete prompts, which they even mentioned as a contribution. However, even after reading the explanation between lines 50 and 61, the contribution of having discrete prompts is unclear. I understand the motivation of having concrete concepts represented as discrete vectors for human understanding, but a continual vector can easily represent the same concept and also be distinct categories and even linearly separable.
    - Furthermore, the implementation of discrete prompts in the proposal must be clarified. While I assume it occurs in the NN look-up or the prompt formation, there is a lack of ablation on this point. For instance, how is prompt usage distributed (Are they uniformly distributed, or are some very specialised)? Are there specialised prompts? Is there a discernible relationship between a 'concept' and a prompt?

**Questions:**

- In line 195, it is mentioned that this selection allows more updates in relevant prompts without disrupting less relevant ones. This idea is very interesting and crucial in CL. However, some questions remain:
    - Concerning this point, how different is a traditional prompt-based approach from the proposal? In both cases, you only update the “relevant” prompt, but the meaning of relevant changes. Or is there something else?
    - The look-up table selects the most relevant prompt; however, we should be able to compose concepts to learn more abstract ideas. Can this proposal be extended to this?
- There is a second training phase when applying the component described in Section 4.3. How many epochs does this new phase have? Could it be an unfair advantage over other methods? For example, how does L2P behave with this component?
- The experiments done in 5.3 concerning the temperature of alpha suggest that a lower value leads to a sharper distribution, which can lead to a more discrete selection; however, a higher value achieves better performance. How can we read this, as it seems contrary to the need for discrete prompts?

**Limitations:**

The authors present some limitations and concerns regarding the proposal. Another limitation that can be added is the intrinsic limitation in the pre-trained models concerning the limitations of the pre-trained data distribution and the computational cost of running these models.

---

> ### Author Rebuttal · Authors · 2024-08-06
>
> We sincerely thank the reviewer for the valuable and insightful comments.
>
> **W1. One of the authors' motivations is learning discrete prompts, which they even mentioned as a contribution. However, even after reading the explanation between lines 50 and 61, the contribution of having discrete prompts is unclear. I understand the motivation of having concrete concepts represented as discrete vectors for human understanding, but a continual vector can easily represent the same concept and also be distinct categories and even linearly separable.**
>
> We would like to clarify that while we refer to them as discrete, the values of these discrete prompts are indeed float (continuous) during optimization. They are distinct from continuous prompts in their conceptual structure and function.
>
> Continuous prompts typically refer to input-conditioned prompts where each input sample has a unique prompt, even if the samples are instances of the same concept. This approach can lead to redundancies in concept representation. In contrast, discrete prompts use a limited set of prompts to represent concepts, which can serve as a more abstract representation of knowledge and align more closely with human cognitive structures. This form of knowledge representation can enhance the model's ability to learn and manage task-specific knowledge efficiently.
>
> **W2. The implementation of discrete prompts in the proposal must be clarified. While I assume it occurs in the NN look-up or the prompt formation, there is a lack of ablation on this point. For instance, how is prompt usage distributed (Are they uniformly distributed, or are some very specialised)? Are there specialised prompts? Is there a discernible relationship between a 'concept' and a prompt?**
>
> Regarding prompt usage distribution, we have observed some prompts are more frequently selected for samples from different tasks while others are less frequently used. Since the prompts are designed to learn task-specific knowledge, each task might not have a universal ‘concept’. However, different tasks might share some features that can be captured by certain prompts. We will further clarify these observations and provide a more detailed analysis in the revised paper.
>
> **Q1. In line 195, it is mentioned that this selection allows more updates in relevant prompts without disrupting less relevant ones. This idea is very interesting and crucial in CL. However, some questions remain:**
> - **How different is a traditional prompt-based approach from the proposal? In both cases, you only update the “relevant” prompt, but the meaning of relevant changes. Or is there something else?**
>
> The discussion in line 195 aims to explain how our design can enable CL. How to effectively select and update relevant prompt(s) is crucial for achieving CL, and the meaning of “relevant” in our context shares the same insight as traditional prompt-based methods.  However, the primary difference lies in whether the prompt selection can be optimized with the task loss.
>
> - **The look-up table selects the most relevant prompt; however, we should be able to compose concepts to learn more abstract ideas. Can this proposal be extended to this?**
>
> One possible way is to design a two-level prompt pool, with each level responsible for a different degree of abstraction. The model can first identify relevant prompts from the pool of lower abstraction. Next, it learns the combination weights to form a more abstract prompt. This newly composed prompt is then matched with a prompt from the higher abstraction pool, and the matched one is used for further processing.
>
> **Q2. There is a second training phase when applying the component described in Section 4.3. How many epochs does this new phase have? Could it be an unfair advantage over other methods? For example, how does L2P behave with this component?**
>
> Ten epochs. This phase stabilizes task knowledge learning by leveraging representation statistics of old data, which can contribute to performance improvements but is not the sole determinant of the final performance.
>
> Following your suggestion, we trained the L2P++ with this component. As can be observed in the following table, L2P++ with representation statistics (“L2P++ V2”) performs better than the original L2P ++ in the three settings on ImageNet-R, but remains inferior to our method. Additionally, we list the performance of VQ-Prompt-S, a simplified version of VQ-Prompt that does not use representation statistics. VQ-Prompt-S still outperforms other methods such as EvoPrompt, demonstrating the effectiveness of our approach.
> ||5-task FAA|5-task CAA|10-task FAA|10-task CAA|20-task FAA|20-task CAA|
> |-|-|-|-|-|-|-|
> |L2P++|70.83$\pm$0.58|78.34$\pm$0.47|69.29$\pm$0.73|78.30$\pm$0.69|65.89$\pm$1.30|77.15$\pm$0.65|
> |L2P++ V2|74.11$\pm$0.08|78.44$\pm$0.63|72.93$\pm$0.27|78.63$\pm$0.80|70.99$\pm$0.26|77.65$\pm$0.79|
> |EvoPrompt|77.16$\pm$0.18|82.22$\pm$0.54|76.83$\pm$0.08|82.09$\pm$0.68|74.41$\pm$0.23|80.96$\pm$1.42|
> |VQ-Prompt-S|78.52$\pm$0.34|82.64$\pm$0.68|78.00$\pm$0.39|82.83$\pm$0.69|76.19$\pm$0.26|81.68$\pm$1.02|
>
> **Q3. The experiments done in 5.3 concerning the temperature of alpha suggest that a lower value leads to a sharper distribution, which can lead to a more discrete selection; however, a higher value achieves better performance. How can we read this, as it seems contrary to the need for discrete prompts?**
>
> This can be understood by considering the balance between discrete selection and robust learning. While a lower temperature leads to a sharper distribution and more discrete selections, it can lead to overly aggressive prompt choices early in training when the model is less fully trained. This can negatively impact the learning process and overall performance, but it does not negate the value of discrete prompts themselves.
>
> **Another limitation.**
>
> Thanks for the insightful suggestion. We will include a discussion of the intrinsic limitations of pre-trained models in the revised paper.

---

> > ### Comment · Reviewer_GsNc · 2024-08-09
> >
> > I thank the author's answers. For now, I will keep my score waiting for comments from the other reviewers.

---

### Official Review · Reviewer_7MCa · 2024-07-12

**Soundness:** 3
**Presentation:** 3
**Contribution:** 3
**Rating:** 5
**Confidence:** 4

**Summary:**

This paper introduces Vector Quantization Prompting (VQ-Prompt), a novel prompt-based method designed to mitigate catastrophic forgetting in the sequential learning scenario of Class Incremental Learning (CIL). VQ-Prompt utilizes Vector Quantization (VQ) to facilitate end-to-end learning with a set of discrete prompts. Initially, it computes the queries for the input images and determines similarity scores between the query and keys, similar to the Learning to Prompt (L2P) approach. This involves using a pretraining image method to generate queries. Subsequently, it aggregates prompts based on these similarity scores, resulting in what is called a continuous prompt. This aggregated prompt is then quantized by selecting the nearest neighbor (NN) from the prompt pool. Then, the pipeline to classify the instances remains constant compared to L2P. To ensure gradient propagation through the key and all prompts, VQ-Prompt employs the straight-through estimator along with similarity scores. Additionally, it incorporates representation statistics to address classification bias on the classifier. The authors demonstrate that VQ-Prompt outperforms state-of-the-art (SOTA) baselines across three datasets: ImageNet-R (5, 10, and 20 tasks), CIFAR-100 (10 tasks), and CUB-200 (10 tasks).

**Strengths:**

This paper presents the following strengths:

1. The paper is well-written and organized. It is easy to follow.
2. The problem studied is highly relevant and valuable.
3. The paper proposes a method that enables end-to-end training for architecture based on prompt learning for CIL and shows that this can also be beneficial in achieving state-of-the-art results in three well-known datasets.

**Weaknesses:**

1. Considering Prompt Learning for Continual Learning (CL) assumes that there is a good and generalizable pretraining that could be transferable (tuned) to the different downstream tasks. Moreover, considering the way that the queries are generated, it also presumes that this pretraining can accurately select prompts for all tasks. I would like to see results on more challenging tasks like out-of-domain. Although the authors perform the evaluation in different scenarios, two of them, ImageNet-R and CIFAR-100, are closely related to ImageNet. For instance, ImageNet-R is derived from ImageNet-1K, and also its classes overlap with ImageNet-1K.
2. Additionally, it is worth noting that the authors use a larger pretraining dataset for the experiments in Table 3, where VQ-Prompt achieves nearly the same performance as HiDe-Prompt. This raises the question of whether this is related to the initialization issue mentioned earlier. I would like to see these results with a different initialization to understand this relationship better.
3. There is no consistency in the evaluation of state-of-the-art (SOTA) methods across all tasks (Table 1, Table 2, Table 3), making comparison and analysis challenging. For instance, HiDe-Prompt shows a significant improvement over CODA-P and almost matches the performance of VQ-Prompt. So, why is it not evaluated on the other tasks? I would like to see how HiDe-Prompt performs on the other tasks presented in Table 1 and Table 2.

**Questions:**

Please refer to the weaknesses section, I left some questions for you.

**Limitations:**

The authors mention a limitation of this method: scaling up the prompt size does not increase performance. However, this is not necessarily a limitation. This could indicate that the model does not need significant parameters to learn the task because it is relatively easy due to its initial knowledge. (Please see point 1 of the weaknesses).

---

> ### Author Rebuttal · Authors · 2024-08-06
>
> We sincerely thank the reviewer for the valuable and insightful comments.
>
> **W1. Considering Prompt Learning for Continual Learning (CL) assumes that there is a good and generalizable pretraining that could be transferable (tuned) to the different downstream tasks. Moreover, considering the way that the queries are generated, it also presumes that this pretraining can accurately select prompts for all tasks. I would like to see results on more challenging tasks like out-of-domain. Although the authors perform the evaluation in different scenarios, two of them, ImageNet-R and CIFAR-100, are closely related to ImageNet. For instance, ImageNet-R is derived from ImageNet-1K, and also its classes overlap with ImageNet-1K.**
>
> Please note that ImageNet-R is widely recognized as a critical benchmark for CL, especially for methods utilizing backbones pre-trained on the ImageNet, as discussed in DualPrompt. This is because ImageNet-R contains artistic renditions of ImageNet categories that are discouraged by the collecting rules of the original ImageNet. CIFAR-100 has also been widely used in previous prompt-based CL methods.
>
> Following your suggestion, we conducted experiments on two new datasets, namely ImageNet-A [1] and VTAB [2]. ImageNet-A contains adversarial images that fool current ImageNet pre-trained classiﬁers, while VTAB includes 19 datasets with diverse classes that do not overlap with ImageNet-1K. For ImageNet-A, we split the 200 classes into 20 tasks. For VTAB, we sample five 10-class datasets from it to construct the cross-domain CIL setting. As shown below, our VQ-Prompt outperforms other SOTA methods such as HiDe-Prompt on these challenging datasets on the FAA metric.
>
>
> ||ImageNet-A|VTAB|
> |-|-|-|
> |HiDe-Prompt|51.67|86.38|
> |VQ-Prompt|52.96|90.46|
>
>
> [1] Natural adversarial examples. CVPR21
>
> [2] A large-scale study of representation learning with the visual task adaptation benchmark. arXiv19
>
> **W2. Additionally, it is worth noting that the authors use a larger pretraining dataset for the experiments in Table 3, where VQ-Prompt achieves nearly the same performance as HiDe-Prompt. This raises the question of whether this is related to the initialization issue mentioned earlier. I would like to see these results with a different initialization to understand this relationship better.**
>
> It is important to clarify that the results in Table 3 for both VQ-Prompt and HiDe-Prompt use the SAME pre-training dataset, i.e., ImageNet-21K, as denoted in the caption. Our method achieves superior performance compared to HiDe-Prompt, particularly in the CAA metric.
>
> Regarding the initialization with different dataset sizes, besides the results of pre-training on ImageNet-21K shown in Table 3, we present additional comparisons with HiDe-Prompt on ImageNet-R with backbones pre-trained on ImageNet-1K. The results for HiDe-Prompt are obtained by running its official repository. As shown in the table below, our VQ-Prompt consistently outperforms HiDe-Prompt in the 5-task, 10-task, and 20-task settings on ImageNet-R.
>
>
> ||5-task FAA|5-task CAA|10-task FAA|10-task CAA|20-task FAA|20-task CAA|
> |-|-|-|-|-|-|-|
> |CODA-P|76.51$\pm$0.38|82.04$\pm$0.54|75.45$\pm$0.56|81.59$\pm$0.82|72.37$\pm$1.19|79.88$\pm$1.06|
> |HiDe-Prompt|76.29$\pm$0.10|78.77$\pm$0.11|76.74$\pm$0.18|78.76$\pm$0.11|76.46$\pm$0.06|78.76$\pm$0.11|
> |VQ-Prompt|79.32$\pm$0.29|82.96$\pm$0.50|78.71$\pm$0.22|83.24$\pm$0.68|78.10$\pm$0.22|82.70$\pm$1.16|
>
>
> Moreover, we have conducted experiments with different initializations originating from various self-supervised pre-training paradigms, as shown in Section 5.2 and Table 4. The findings demonstrate that VQ-Prompt consistently achieves competitive performance across different initializations, highlighting its robustness and adaptability.
>
> **W3. There is no consistency in the evaluation of state-of-the-art (SOTA) methods across all tasks (Table 1, Table 2, Table 3), making comparison and analysis challenging. For instance, HiDe-Prompt shows a significant improvement over CODA-P and almost matches the performance of VQ-Prompt. So, why is it not evaluated on the other tasks? I would like to see how HiDe-Prompt performs on the other tasks presented in Table 1 and Table 2.**
>
> We would like to clarify that results in Tables 1, 2 and 3 are taken from previous publications, which utilize backbones pre-trained on different datasets. For example, HiDe-Prompt in Table 3 uses weights pre-trained on ImageNet-21K, whereas methods in Tables 1 and 2 are pre-trained on ImageNet-1K. This difference in pre-training datasets impedes direct comparisons across these tables.
>
> Following your suggestion, we ran the official code of HiDe-Prompt and obtained its performance with a backbone pre-trained on ImageNet-1K. The results on ImageNet-R are shown in the **Table in the response to W2**. As can be observed, HiDe-Prompt achieves inferior results compared to our method on ImageNet-R. Unfortunately, we could not provide results for CIFAR-100 because we failed to obtain reasonable results with the settings in the paper (bs=128 and lr=0.005). We found that the actual learning rate changes with the batch size. We are exploring other training settings.
>
> **The authors mention a limitation of this method: scaling up the prompt size does not increase performance. However, this is not necessarily a limitation. This could indicate that the model does not need significant parameters to learn the task because it is relatively easy due to its initial knowledge. (Please see point 1 of the weaknesses).**
>
> We agree with the reviewer's perspective that this could also be seen as a strength of our method. Our model does not require significant parameters to learn tasks due to its effective end-to-end knowledge learning achieved by integrating VQ into the prompt-based framework. We will revise our manuscript to provide a more in-depth discussion regarding the scalability of our approach.

---

> > ### Comment · Reviewer_7MCa · 2024-08-12
> > **Rebuttal Response**
> >
> > I really appreciate the author's effort to address all my concerns. The new results provided by the authors show that the model can work in challenging scenarios for pretraining knowledge. Therefore, I am considering increasing my score.
> >
> > Best,
> > Reviewer 7MCa

---

### Official Review · Reviewer_xyoq · 2024-07-15

**Soundness:** 3
**Presentation:** 3
**Contribution:** 2
**Rating:** 5
**Confidence:** 4

**Summary:**

Prompt-based continual learning has emerged to address catastrophic forgetting in sequential task learning by using a pre-trained Vision Transformer (ViT) enhanced with learnable "prompts." These prompts contain task-specific knowledge and guide the model in generating task-relevant outputs. During inference, the most suitable prompt is selected based on the input image. However, current methods face challenges due to non-differentiable indexing, leading to suboptimal prompt selection and performance issues. Discrete prompts offer better task knowledge representation and prevent interference among different tasks. This paper introduces Vector Quantization Prompting (VQ-Prompt) for continual learning, which optimizes prompts using task loss while maintaining their discrete properties. The method generates a continuous prompt and replaces it with the nearest match from a prompt pool, using gradient estimation to handle non-differentiability and additional regularization to refine the prompt pool. Representation statistics are used to mitigate classification bias towards previous tasks, improving continual learning performance.

**Strengths:**

This paper develops a SoTA prompt-based continual learning method that improves prompt design with a widely used idea of vector quantization. The proposed methodology is sound and rational in design and outperforms other recent strong baselines.

The paper is easy to follow and provides a meaningful analysis to better understand the key components of the proposed approach.

**Weaknesses:**

The performance improvements are meaningful, but not significant - particularly on specific tasks, such as CIFAR-100 and CUB-200 benchmarks.

The technical contribution of the proposed method is not solid. Indeed I didn't find unique insightful ideas/analyses/novelty that I can get only from this paper.  Most technical components used in this work are somewhat general in vector quantization (e.g., NN look-up and straight-through gradient estimation) and continual learning literature (e.g.,post-adjustment of representation based on statistics). And this proposed method is more like a good combination of vector quantization idea for prompt design during continual learning.

The question of "Why is the vector quantization idea more beneficial for prompt-based continual learning against other baselines" is not sufficiently discussed. It achieves competitive performance with a smaller prompt length than other continuous-prompt methods. However, the paper does not provide any analysis/evidence as to why this happens, but only gives typical ablation studies.


In Table 4, the proposed method seems to have pros and cons for different self-supervised pre-training paradigms. It shows better cumulative accuracy (CAA) for on-par final accuracy (FAA); in my understanding, this means the proposed method shows a higher degree of forgetting compared to baselines, while showing better adaptation performance to new tasks.

**Questions:**

-

**Limitations:**

The authors addressed the limitations and broader impacts.

---

> ### Author Rebuttal · Authors · 2024-08-06
>
> We sincerely thank the reviewer for the valuable and insightful comments.
>
> **W1. The performance improvements are meaningful, but not significant - particularly on specific tasks, such as CIFAR-100 and CUB-200 benchmarks.**
>
> We respectfully disagree with the assessment that the improvements are not significant. As demonstrated in Table 1, our method achieves a relative improvement of 2.80%, 2.45%, and 4.96% over the second-best method, EvoPrompt, on the 5-task, 10-task, and 20-task settings of ImageNet-R using the FAA metric. ImageNet-R is widely recognized as a critical benchmark for continual learning, especially for methods utilizing backbones pre-trained on the ImageNet dataset, as discussed in DualPrompt.
>
> Additionally, while our improvements on CIFAR-100 (Table 2) and CUB-200 (Table 3) are not as large as those on ImageNet-R, they are still considerable. On CUB-200, although we slightly trail behind SLCA in the CAA metric, it is important to note that SLCA does not freeze the backbone and employs a higher number of trainable parameters, allowing for better adaptation to the downstream task of fine-grained classification.
>
> The differences in performance improvements might be attributed to the inherent characteristics of the datasets. CIFAR-100 and CUB-200 have data distributions that are closer to the original ImageNet distribution compared to ImageNet-R, which contains artistic renditions of ImageNet categories that are discouraged by the collecting rules of the original ImageNet. This closer alignment with ImageNet may reduce the relative improvement margins on CIFAR-100 and CUB-200.
>
> **W2. The technical contribution of the proposed method is not solid. Indeed I didn't find unique insightful ideas/analyses/novelty that I can get only from this paper. Most technical components used in this work are somewhat general in vector quantization (e.g., NN look-up and straight-through gradient estimation) and continual learning literature (e.g.,post-adjustment of representation based on statistics). And this proposed method is more like a good combination of vector quantization idea for prompt design during continual learning.**
>
> Thanks for the comment. Our method is motivated by identifying a clear disadvantage in current prompt-based methods, i.e., the unoptimized selection of prompts. We propose a simple yet effective approach to address this issue by integrating vector quantization with prompt-based continual learning, which has not been explored before. While our technical components are based on established principles in vector quantization and continual learning, the innovation lies in the unique application and combination of these techniques to solve the specific problem of prompt selection. By incorporating vector quantization, we optimize the prompt selection process with task loss to achieve a more effective abstraction of task knowledge, which is crucial for continual learning. Experimental results demonstrate the effectiveness of our approach. Thus, while we build upon existing techniques, our work provides a fresh perspective and a practical solution to a challenge in prompt-based continual learning.
>
> **W3. The question of "Why is the vector quantization idea more beneficial for prompt-based continual learning against other baselines" is not sufficiently discussed. It achieves competitive performance with a smaller prompt length than other continuous-prompt methods. However, the paper does not provide any analysis/evidence as to why this happens, but only gives typical ablation studies.**
>
> Thanks for the insightful comment. The use of vector quantization (VQ) in our method provides several key benefits for prompt-based continual learning. First, VQ allows us to encode task knowledge into discrete prompts, which provide a more compact representation compared to continuous prompts, as discussed between lines 50 and 61 of the manuscript. This discrete nature helps in capturing essential task-specific features with necessary abstraction compared with continuous representations. Second, by integrating VQ into the prompt-based framework, we enable the optimization of prompt selection with task loss. This end-to-end optimization ensures that the selected prompts are highly relevant to the task at hand, thereby enhancing the task knowledge learning of the prompts. In summary, we can use a smaller prompt length to achieve competitive performance due to effective end-to-end knowledge learning. We will extend our analysis in the revised manuscript by providing detailed explanations.
>
> **W4. In Table 4, the proposed method seems to have pros and cons for different self-supervised pre-training paradigms. It shows better cumulative accuracy (CAA) for on-par final accuracy (FAA); in my understanding, this means the proposed method shows a higher degree of forgetting compared to baselines, while showing better adaptation performance to new tasks.**
>
> Thanks for the detailed observation. The higher CAA suggests that VQ-Prompt is more efficient in leveraging past knowledge to aid current tasks, even though it might exhibit a higher degree of forgetting compared to some baselines. This trade-off between adaptation to new tasks and forgetting of previous ones is a common challenge in continual learning. With self-supervised pre-training paradigms, our approach tends to prioritize adaptability to new tasks to ensure that the model remains relevant and effective in dynamic environments. In our revised manuscript, we will provide additional analysis to elucidate this trade-off and further highlight the strengths and limitations of VQ-Prompt in various self-supervised pre-training scenarios.

---

### Decision · Program_Chairs · 2024-09-25

**Decision:**

Accept (poster)

**Comment:**

I agree with the unanimous decision of the reviewers to accept the paper. However, there has been some discussion regarding its presentation. I strongly encourage the authors to incorporate the reviewers' feedback and include the new results in the camera-ready version of the paper.